# Rhabdomyolysis in the Era of Next-Generation Sequencing: Selecting Patients with a High Likelihood of a Genetic Susceptibility Using ‘RHABDO’ Features

**DOI:** 10.3390/genes16090995

**Published:** 2025-08-25

**Authors:** Nick Kruijt, Sanne A. J. H. van de Camp, Jasper J. Kramer, Luuk R. van den Bersselaar, Meyke Schouten, Thatjana Gardeitchik, Heinz Jungbluth, Salman Bhai, Anneke J. van der Kooi, Erik-Jan Kamsteeg, Nicol C. Voermans

**Affiliations:** 1Department of Neurology, Radboudumc Research Institute for Medical Innovation, Donders Institute for Brain, Cognition and Behaviour, Radboud University Medical Center, 6525 XZ Nijmegen, The Netherlands; sanne.vandecamp@radboudumc.nl (S.A.J.H.v.d.C.); jasper.kramer@radboudumc.nl (J.J.K.); luuk.vandenbersselaar@radboudumc.nl (L.R.v.d.B.); nicol.voermans@radboudumc.nl (N.C.V.); 2Malignant Hyperthermia Investigation Unit, Department of Anesthesiology, Canisius Wilhelmina Hospital, 6532 SZ Nijmegen, The Netherlands; 3Department of Anesthesiology, Pain and Palliative Medicine, Radboud University Medical Center, 6500 HB Nijmegen, The Netherlands; 4Department of Human Genetics, Radboud University Medical Center, 6525 GA Nijmegen, The Netherlands; meyke.schouten@radboudumc.nl (M.S.); thatjana.gardeitchik@radboudumc.nl (T.G.); erik-jan.kamsteeg@radboudumc.nl (E.-J.K.); 5Muscle Signalling Section, Randall Centre for Cell and Molecular Biophysics, Faculty of Life Sciences and Medicine (FoLSM), King’s College London, London SE1 1UL, UK; heinz.jungbluth@gstt.nhs.uk; 6Department of Paediatric Neurology, Neuromuscular Service, Evelina London Children’s Hospital, Guy’s and St Thomas’ Hospital NHS Foundation Trust, London SE1 7EH, UK; 7Department of Neurology, University of Texas Southwestern Medical Center, Dallas, TX 75390, USA; salman.bhai@utsouthwestern.edu; 8Neuromuscular Center, Institute for Exercise and Environmental Medicine, Texas Health Presbyterian, Dallas, TX 75231, USA; 9Department of Neurology, Amsterdam University Medical Center, Amsterdam Neuroscience Institute, University of Amsterdam, 1081 HV Amsterdam, The Netherlands; a.j.kooi@amsterdamumc.nl

**Keywords:** rhabdomyolysis, neuromuscular disorders, genetics, genetic predisposition to disease, next-generation sequencing

## Abstract

Background/Objectives: Rhabdomyolysis is a potentially life-threatening condition characterized by acute skeletal muscle breakdown. In addition to well-described external triggers, a genetic contribution is increasingly recognized. We aimed to (I) review the genetic diagnostic approach of rhabdomyolysis, (II) evaluate the clinical characteristics indicative of a genetic susceptibility with the ‘RHABDO’ acronym, and (III) assess the predictive value of the presence RHABDO features for identifying genetic variants. Methods: In this retrospective two-center study, 122 patients underwent genetic testing for rhabdomyolysis since the introduction of whole exome sequencing (WES) in 2013. The presence of RHABDO features was compared between those with (likely) pathogenic variants and those with benign or no identified variants or variants of uncertain significance. Results: The testing methods included panel-based WES (82%), Sanger sequencing (49%), and full WES (24%), of which 52 patients (43%) underwent multiple methods. A (likely) pathogenic variant was identified in 13 patients (11%), in all of whom ≥2 RHABDO features were present. The positive predictive value for ≥2 features was 14%, while the negative predictive value was 100%. Conclusions: These results highlight the relevance of WES in further elucidating the genetic basis of rhabdomyolysis and demonstrated that RHABDO is a valuable tool for selecting patients who should undergo genetic testing.

## 1. Introduction

Rhabdomyolysis is a life-threatening condition involving the rapid dissolution of skeletal muscle in response to a variety of triggers, clinically characterized by a sudden marked rise and subsequent fall in serum creatine kinase (CK) levels [1]. The wide range of complications (e.g., acute renal failure, cardiac arrhythmias, compartment syndrome, disseminated intravascular coagulation) emphasizes the clinical relevance of rhabdomyolysis across many medical disciplines [2,3,4,5]. Retrospective cohort studies focusing on rhabdomyolysis in hospitalized patients in the pre-next-generation sequencing (NGS) era have concentrated particularly on external triggers as the major cause for a rhabdomyolysis event. These studies identified exogenous toxins (e.g., illicit drugs, alcohol), direct muscle trauma, infections, and strenuous exercise as among the most common triggers [3,6,7,8]. However, a genetic contribution is increasingly recognized, with over 30 genes currently associated with an increased rhabdomyolysis susceptibility [7]. Due to the more wide-spread availability of NGS, more supportive evidence for a potentially causative role of genetic defects is expected [7,9,10,11,12]. However, a key challenge in the diagnostic approach lies in the consideration of which patients require diagnostic genetic screening following a rhabdomyolysis episode to identify an otherwise pauci- or asymptomatic neuromuscular or metabolic disorder.

To inform this question, we have previously proposed the acronym ‘**RHABDO**’ [2,10,13]. Based on this acronym, we recommend considering a genetic cause of rhabdomyolysis in patients with one or more of the following features: **R**ecurrent episodes; **H**yperCKaemia persisting 8 weeks after the event; **A**ccustomed exercise—the intensity of the exercise cannot sufficiently explain the rhabdomyolysis event; **B**lood CK > 50× the upper limit of normal (ULN) or >10,000 IU/L in female Caucasian patients; **D**rugs/medication and other exogenous triggers are insufficient to explain the event; and **O**ther affected family members or other exertional symptoms (e.g., severe muscle cramps or swelling). The acronym was based on a literature review and on our clinical experience, and, therefore, reached the evidence level of expert opinion. The relevance of RHABDO is further supported by a recent European NeuroMuscular Center (ENMC) workshop involving 21 physicians and researchers from 12 different countries, which emphasized the need for coordinated research in this area [14]. In addition, the use of the acronym has been adopted in previous studies by other experts in the field [10,13,15].

Our previous retrospective hospital-based study, which included a total of 1302 patients presenting with an acute CK level exceeding 2000 IU/L between 2013 and 2019, evaluated the exogenous triggers and results of laboratory testing [7]. In the majority of patients with unexplained rhabdomyolysis (*n* = 193/277; 70%), one or more of the RHABDO features was present, and a (likely) pathogenic variant was identified in 72 patients (*n* = 72/193, 37%). In 121 patients who fulfilled the ≥1 RHABDO features criterium, no genetic variant was identified; however, panel-based WES had not been performed in the majority of patients (*n* = 82). Considering the 37% yield in the cohort of patients with ≥1 RHABDO features who underwent NGS, it is likely that a proportion of the untested patients with a RHABDO feature also carry a (likely) pathogenic genetic variant. These findings prompted a more systematic evaluation of the predictive value of the RHABDO features in a cohort of patients who had undergone genetic testing.

In the present study, we aimed to (I) review the current genetic diagnostic approach and genetic test results in a cohort of patients with rhabdomyolysis who underwent genetic testing after the introduction of NGS in the Netherlands; (II) assess the presence of RHABDO features in this cohort; and (III) evaluate the predictive value of the RHABDO features for the presence of (likely) pathogenic variants. We expect that our results may help to formulate strategies to further elucidate genetic causes and to provide recommendations for clinicians on when to consider genetic testing in patients presenting with rhabdomyolysis.

## 2. Materials and Methods

### 2.1. Identification of Patients

Patients were identified through a systematic search for ‘rhabdomyolysis’ in the database of the diagnostic genetics laboratory at the Radboud University Medical Center, Nijmegen, and the Amsterdam University Medical Center, the Netherlands. Patients were eligible if the clinical information on the genetic testing form included ‘rhabdomyolysis’ and a genetic test had been performed since the introduction of whole exome sequencing (WES) in July 2013 until November 2021. The study was approved by the ethical committee of the Radboud University Medical Center (CMO numbers 2017-4022, 2019-5217, and 2020-6649).

### 2.2. Data Collection

Data were extracted from medical records, including information on (I) clinical characteristics (i.e., sex, age, medical history, and presence of RHABDO features); (II) potential triggers (i.e., exercise, fever/infection, dehydration, illicit drug abuse, excessive alcohol intake, prolonged immobilization, medication, hypothyroidism, vitamin D deficiency or direct muscle trauma); (III) clinical signs of myoglobinuria or symptoms persisting for up to five days after the event (i.e., myalgia, muscle cramps, weakness, stiffness, or swelling); and (IV) peak serum CK levels. The specific type of genetic testing performed was documented (i.e., targeted gene testing and/or panel-based WES or full WES). The most recent panel versions used in the Radboud University Medical Center and Amsterdam University Medical Centers were DG3.8.1 and NGS neuromuscular disorders v3, available online [16,17]. Variant pathogenicity was assessed in accordance with the standards and guidelines for the interpretation of sequence variants by The American College of Medical Genetics and Genomics (ACMG) and the Dutch Society of Clinical Genetic Laboratory Specialists (VKGL) [18]. The corresponding rsIDs were obtained by cross-referencing the variants with the dbSNP (Build 153) [19]. In order to describe the data, patients were categorized based on the pathogenicity of the identified variant: (I) presence of a (likely) pathogenic variant, or (II) presence of a variant of uncertain significance (VUS), benign variant(s), or no genetic variant.

### 2.3. Statistical Analysis

Statistical analysis was performed using the Statistical Package for the Social Sciences (SPSS v24; IBM Corp., Armonk, NY, USA). The two groups were compared to assess differences in clinical characteristics; continuous variables were tested for normal distribution using the Shapiro–Wilk test and presented as mean ± standard deviation in case of normal distribution, and median [Interquartile range (IQR)] in case of non-normal distribution. Categorical data were compared using the Chi-squared test or Fisher’s exact test. Continuous variables were compared using an independent two-sample t-test, or Mann–Whitney U-test in case of non-normal distribution. Sensitivity and specificity for detecting a (likely) pathogenic genetic defect were calculated based on a cut-off of fewer than two RHABDO features. The level of significance was set at *p* < 0.05.

## 3. Results

A total of 122 patients was included (Table 1), with 87 male (71%) and 35 female (29%) patients. The median age at the time of the first rhabdomyolysis event was 29 years (22–43 years). The median age at the time of the most recent genetic test was 38 years (27–51 years). The median peak CK was 17,000 IU/L (9072–39,250 IU/L).

### 3.1. Genetic Testing

Sanger sequencing was performed in 60 patients (49%), panel-based WES in 100 patients (82%), and full WES in 29 patients (24%). More than half of the patients who initially underwent Sanger sequencing were re-evaluated at a later timepoint using panel-based or full WES (*n* = 38/60, 63%). Additionally, in 6 of the 100 patients who underwent WES, reanalysis with an updated WES panel was performed at a later timepoint.

In 41 patients (33%), one or multiple genetic variants were identified. Of those, 13 patients (11%) carried a (likely) pathogenic variant (Table 2), while 10 patients (8%) had one or multiple VUS (Table 3). Additionally, 18 patients (15%) carried variants that were initially classified as VUS, which were reclassified as benign or likely benign upon re-evaluation.

Out of thirteen patients with a (likely) pathogenic variant, 11 variants were identified through panel-based WES. Of those, six patients had initially undergone Sanger sequencing, which had not detected any pathogenic variants. Two pathogenic variants were identified with Sanger sequencing and were not subsequently tested using WES. In six patients, WES was repeated at a later timepoint using an updated gene panel, which revealed a pathogenic variant in three patients. The 13 (likely) pathogenic variants were located in ten different genes: *ACADVL*, *ANO5*, *CPT2*, *DMD*, *DYSF*, *FKRP*, *PGAM2*, *PGM1*, *PYGM* and *RYR1*. Variants in *CPT2* and *RYR1* were identified in two and three patients, respectively. In ten patients, multiple VUSs were detected, which were located in seven different genes: *ATP7A*, *DES*, *DYSF*, *HSPB8*, *TTN*, *SMCHD1*, *RYR1*. Variants in *RYR1* were present in four patients.

### 3.2. RHABDO Features

In 119 patients (98%), ≥1 RHABDO feature was present, and 92 patients (75%) fulfilled the criteria of two or more features. The most commonly reported feature was a serum CK level > 50× ULN (*n* = 91, 75%), followed by recurrent episodes of rhabdomyolysis (*n* = 77, 63%). Furthermore, the most frequently reported exogenous trigger for rhabdomyolysis was exercise (72%, *n* = 88), followed by fever (22%, *n* = 27). In nine (7%) patients, no clearly identifiable trigger for the rhabdomyolysis event was reported. No differences were observed in the prevalence of each individual RHABDO feature between the two groups. However, the number of RHABDO features present was higher in the group with a (likely) pathogenic variant (*p* = 0.03). All patients with a (likely) pathogenic variant had at least two reported RHABDO features; the positive predictive value of the presence of ≥2 RHABDO features for detecting a (likely) pathogenic variant was 14%, whereas the negative predictive value of ≤1 RHABDO feature for the absence of a pathogenic variant was 100%.

## 4. Discussion

In this two-center retrospective study, we describe a cohort of 122 patients with rhabdomyolysis who underwent genetic testing since the introduction of WES in 2013. A (likely) pathogenic variant was identified in 11% of patients, whereas a VUS was detected in 8% of patients. The positive predictive value of ≥2 RHABDO features was 14%, and the negative predictive value was 100%. In three out of six patients, repeating WES using an updated gene panel revealed a pathogenic variant after reanalysis.

Previous studies investigating genetic testing outcomes in patients with elevated serum creatine kinase (CK) levels have reported diagnostic yields ranging from 15% to 20%. Some reports have indicated diagnostic rates as high as 50% [20,21,22]. In comparison, the diagnostic yield of 11% observed in the present study may appear relatively low. This discrepancy is likely attributable to several factors. First, we selected patients with a confirmed rhabdomyolysis episode based on the characteristic rise and fall in serum CK levels. In contrast, previous studies often included patients based on a single CK measurement that was not reassessed, despite the importance of the typical CK trajectory in distinguishing persistent hyperCKemia from rhabdomyolysis [23]. This difference in patient selection may increase the likelihood of identifying genetic conditions associated with permanent hyperCKemia, such as certain muscular dystrophies, including limb-girdle muscular dystrophy type R12 (LGMD R12), which can initially present without muscle weakness. Next, many patients included in our study were referred to our tertiary care center after prior diagnostic evaluations had failed to identify an underlying cause, which may introduce referral bias.

The PPV of the presence of ≥2 RHABDO features for detecting a (likely) pathogenic variant by performing panel-based WES was 14%. Given that a substantial proportion of patients presenting with rhabdomyolysis remain genetically undiagnosed, we consider the diagnostic yield observed in this selected cohort to be clinically valuable. It supports the advice in recent international consensus to consider genetic testing as a first-tier diagnostic test. At the same time, the use of panel-based tests has its limitations. For example, the technology is not able to analyze copy number variations (CNVs). Therefore, an ongoing research project in Australia aims to comprehensively assess patients by using whole genome-based diagnostics to identify variants and genes that remained undetected using existing genetic tests. CNV accounted for 12% of the diagnoses (unpublished data) [24].

Many patients who were tested through WES had previously undergone targeted Sanger sequencing, likely reflecting the inclusion period, which commenced immediately after the introduction of WES in November 2013 and lasted until 2021. Sanger sequencing was initially preferred to establish a genetic diagnosis, and during the course of the observation period, panel-based WES and full WES became less expensive and more accessible. Research by Westra et al. showed that WES is an effective way to detect genetic causes of neuromuscular disorders in the Netherlands, identifying disease-causing variants in 19% of patients [12], which is comparable to the yield of genetic variants in the current study. The WES muscle disorders gene panels that were used in both centers were updated approximately three times per year. Therefore, different versions of panel testing were used across patients in the current cohort [16,17]. The number of tested genes varied from 117 in the first panel version to up to 253 genes in the current panel versions. The continuously expanding list of genes included should prompt clinicians to reanalyze patients with unexplained rhabdomyolysis at a later timepoint after NGS gene panels have been updated. This is highlighted by the finding that the reanalysis of six patients in our cohort revealed a previously uncovered variant in three patients.

The most frequently reported triggers contributing to rhabdomyolysis events were exercise (72%), fever/infection (22%), and/or medication (18%). These findings differ from those of previous studies, in which the most frequently reported triggers included direct muscle trauma, use of illicit drugs, or prolonged immobilization, which occurs particularly in the elderly population [3,6]. The discrepancy compared to our study can be explained by the fact that our cohort involved patients in whom genetic testing was requested, which covers only a subpopulation of all patients with rhabdomyolysis. Furthermore, the majority of our patients were outpatients, whereas previous studies primarily focused on hospitalized patients or those seen in the emergency department [3,6,13].

The current approach has several limitations that should be taken into consideration. The study was conducted at two tertiary referral centers for neuromuscular disorders: Radboud University Medical Center and Amsterdam University Medical Center. Both institutions fulfill a national tertiary role in the diagnostic evaluation of complex patients of which a proportion was referred from secondary centers that were unable to identify an underlying cause of the rhabdomyolysis event. As a result, referral patterns may have introduced a certain degree of bias, with an overrepresentation of cases of particular complexity. Nonetheless, with the growing accessibility of NGS in recent years, the results of this study are likely to be relevant to a broader group of clinicians in other institutions other than specialized tertiary centers. Furthermore, identifying the presence of RHABDO features was limited by the retrospective nature of the study, which carries potential bias due to incomplete or missing reported clinical features, and inconsistency in the follow-up of laboratory results; this was particularly evident in CK values, which were often measured at a single timepoint. Ideally, CK values should be followed-up until normalization to formally confirm a diagnosis of rhabdomyolysis. Additionally, only a relatively small number of patients were found to carry a (likely) pathogenic variant, making it difficult to reliably assess phenotypic differences between patients with a (likely) pathogenic variant and those with a VUS. In this study, no statistically significant differences were found between the two groups.

The current study provides valuable insights in the diagnostic approach for identifying individuals with a potential genetic cause of rhabdomyolysis [14,25]. Future studies are needed to validate the predictive value of RHABDO features and assess the relative predictive value of individual RHABDO features. A prospective, multicenter study including both tertiary and local centers would be particularly useful. However, the low prevalence of unexplained rhabdomyolysis poses a considerable challenge. Therefore, collaborative efforts across multiple centers using a standardized protocol to collect data on clinical characteristics would be essential to further assess the clinical utility of the RHABDO features in a clinical setting.

## 5. Conclusions

This study demonstrates that, in the Netherlands, nearly all patients undergoing genetic testing for rhabdomyolysis reported at least one RHABDO feature. A likely genetic cause was identified in 11% of the cohort. Notably, all patients carrying a (likely) pathogenic variant had ≥2 RHABDO features, with a positive predictive value of 14% for patients with ≥2 RHABDO features. In addition, no genetic variants were found in patients without RHABDO features. An international collaboration supported by standardized data collection would be the most effective approach to further evaluate the clinical value of the RHABDO features.

## Figures and Tables

**Table 1 genes-16-00995-t001:** Characteristics of patients with a likely pathogenic or pathogenic variant compared to patients with no pathogenic variant or a variant of uncertain significance.

	Proven or Likely Pathogenicity (*n* = 13) *n* (%)				No Genetic Variant or VUS (*n* =1 09) *n* (%)				Total (*n* = 122) *n* (%)				*p*-Value
**Sex**													
Male	12 (92)				75 (69)				87 (71)				0.11
Female	1 (8)				34 (31)				35 (29)				
**Age**													
At first event (years)	25 (18–33)				30 (22–43)				29 (22–43)				0.27
At last genetic test (years)	29 (20–54)				39 (28–50)				38 (27–50)				0.43
**Presumed underlying cause**													
Muscular dystrophy	3 (23)				1 (1)				4 (3)				
Metabolic/mitochondrial	5 (39)				1 (1)				6 (5)				
*RYR1*-related	2 (15)				1 (1)				3 (2)				
Other inherited	-				-				-				
Heat illness	-				5 (5)				5 (4)				
Medication	-				-				-				
SS/NMS	-				1 (1)				1 (1)				
Other ADR	-				-				-				
Unknown	3 (23)				100 (92)				103 (84)				
**RHABDO features**	**Yes**	**No**		**NR**	**Yes**	**No**		**NR**	**Yes**	**No**		**NR**	
Recurrent	11 (85)	1 (8)		1 (8)	66 (61)	34 (31)		9 (8)	77 (63)	35 (29)		10 (8)	0.13
HyperCKaemia > 8 weeks	5 (39)	2 (15)		6 (46)	20 (18)	23 (21)		66 (61)	25 (20)	25 (20)		72 (59)	0.42
Accustomed exercise	8 (62)	-		5 (38)	39 (36)	6 (6)		64 (59)	47 (39)	6 (5)		69 (57)	0.27
Blood CK > 50× ULN	10 (77)	3 (23)		-	81 (74)	26 (24)		2 (2)	91 (75)	29 (24)		2 (2)	0.58
Drugs/medication	2 (15)	-		11 (85)	14 (13)	1 (1)		94 (86)	16 (13)	1 (1)		105 (86)	0.59
Other family members	3 (23)	10 (77)		-	22 (20)	78 (72)		9 (8)	25 (21)	88 (72)		9 (7)	0.32
**Number of RHABDO features**													0.03 ^†^
0	-				3 (3)				3 (2)				
1	-				27 (25)				27 (22)				
2	5 (39)				33 (30)				38 (31)				
3	5 (39)				35 (32)				40 (33)				
4	1 (8)				11 (10)				12 (10)				
5	2 (15)				-				2 (2)				
**Peak CK (IU/L)**	11,000 (5895–40,500)				17,100 (9358–39,500)				17,000 (9072–39,250)				0.51
**Genetic test performed**	**Yes**		**No**		**Yes**		**No**		**Yes**		**No**		
Sanger	7 (54)		6 (46)		53 (49)		56 (51)		60 (49)		62 (51)		
WES neuromuscular panel	11 (85)		2 (15)		89 (82)		20 (18)		100 (82)		22 (18)		
WES open exome	-		13 (100)		29 (27)		80 (73)		29 (24)		93 (76)		

NR = not reported, ADR = adverse drug reaction, CK = creatine kinase, NMS = neuroleptic malignant syndrome, SS = serotonin syndrome, VUS = variant of uncertain significance, WES = whole exome sequencing. ^†^ Difference in the number of present RHABDO feature irrespective of which feature out of the six is considered.

**Table 2 genes-16-00995-t002:** Genotypes and phenotypes of the 13 individual patients with a likely pathogenic or pathogenic variant.

Patient ID	Gene	Phenotype (rsID)	RHABDO-Criteria	Zygosity	Variant	rsID	Classification
1	*ACADVL*	VLCAD deficiency	R H	Het	Chr17(GRCh37):g.7125494A>C; NM_001270447.2:c.822-2A>C (r.spl?) Chr17(GRCh37):g.7124899G>A; NM_001270447.2:c.589G>A (p.Val197Met)	rs398123092 rs369560930	P LP
2	*ANO5*	LGMD R12	R H A B O	Het Het	Chr11(GRCh37):g.22242653dup; NM_213599.3:c.191dup (p.Asn64fs) Chr11(GRCh37):g.22283777T>C; NM_213599.3:c.1733T>C (p.Phe578Ser)	rs137854521 rs137854526	P LP
3	*CPT2*	CPTII deficiency	R H A B O	Hom	Chr1(GRCh37):g.53668099C>T; NM_000098.3:c.338C>T (p.Ser113Leu)	rs74315294	P
4	*CPT2*	CPTII deficiency	H B O	Hom	Chr1(GRCh37):g.53668099C>T; NM_000098.2:c.338C>T (p.Ser113Leu)	rs74315294	P
5	*DMD*	Becker muscular dystrophy	R B	Hem	ChrX(GRCh37):g.33229421C>T; NM_004006.3:c.9G>A (p.Trp3 *)	rs398122853	LP
6	*FKRP*	LGMD R13	H A B	Hom	Chr19(GRCh37):g.47259533C>A; NM_024301.5:c.826C>A (p.Leu276Ile)	rs28937900	P
7	*PGAM2*	GSD10	R A	Het	Chr7(GRCh37):g.44105109del; NM_000290.4:c.20del (p.Val7fs) Chr7(GRCh37):g.44105101G>A; NM_000290.4:c.28C>T (p.Arg10Trp)	rs764567774 rs529371882	P LP
8	*PGM1*	Congenital Disorder of Glycosylation Type It	R B D	Het	Chr1(GRCh37):g.64102019G>C; NM_002633.3:c.988G>C (p.Gly330Arg)	rs777164338	P
9	*PYGM*	GSD5 (McArdle’s disease)	R B	Hom	Chr11(GRCh37):g.64527223G>A; NM_005609.4:c.148C>T (p.Arg50 *)	rs116987552	P
10	*RYR1*	MH susceptibility	R A B	Het	Chr19(GRCh37):g.39071043G>A; NM_000540.3:c.14545G>A (p.Val4849Ile)	rs118192168	LP
11	*RYR1*	MH susceptibility	R A B	Het	Chr19(GRCh37):g.39076780C>T; NM_000540.3:c.14918C>T (p.Pro4973Leu)	rs146876145	LP
12	*DYSF*	Dysferlinopathy	R A	Het	Chr2(GRCh37):g.71742846C>T; NM_003494.3:c.757C>T; (p.Arg253Trp)	rs146876145	LP
13	*RYR1*	MH susceptibility	R A B D	Het	Chr19(GRCh37):g.38973933A>G; NM_000540.3:c.4711A>G; (p.Ile1571Val) Chr19(GRCh37):g.39009932G>A; NM_000540.3:c.10097G>A; (p.Arg3366His) Chr19(GRCh37):g.39034191A>G; NM_000540.3:c.11798A>G; (p.Tyr3933Cys)	rs146429605 rs137932199 rs147136339	LP

CPTII = carnitine palmitoyltransferase II, GSD = glycogen storage disease, Hom = homozygous, Het = heterozygous, LGMD = limb girdle muscle dystrophy, MH = malignant hyperthermia, VLCAD = very-long-chain acyl-CoA dehydrogenase.

**Table 3 genes-16-00995-t003:** Genotypes of 10 patients in whom a VUS was identified.

Patient ID	Gene	RHABDO-Criteria	Zygosity	rsID	Variant
1	*DES*	R H A B	Het	rs1029457073	Chr2(GRCh37):g.220286195G>A; NM_001927.4:c.1157G>A (p.Arg386His)
2	*TTN*	A	Het	rs202234172	Chr2(GRCh37):g.179554624C>T; NM_133378.4:c.28031-1G>A; r.(spl)
3	*ATP7A*	R	Hemi	-	ChrX(GRCh37):g.77245316T>A; NM_000052.5:c.1198T>A (p.Ser400Thr)
4	*HSPB8*	H B	Het	rs1179250162	Chr12(GRCh37):g.119624859G>A; NM_014365.3:c.397G>A (p.Gly133Ser)
5	*DYSF*	R A	Het	rs139754493 rs115407852	Chr2(GRCh37):g.71780300A>G; NM_003494.4:c.1966A>G (p.Lys656Glu) Chr2(GRCh37):g.71908183G>A; NM_003494.4:c.5999G>A (p.Arg2000Gln)
6	*SMCHD1*	R A B	Het	rs746741499	Chr18(GRCh37):g.2751336G>T; NM_015295.3:c.4226G>T (p.Arg1409Leu)
7	*RYR1*	R	Het	-	Chr19(GRCh37):g.38958450C>A; NM_000540.3:c.3379C>A (p.Arg1127Ser)
8	*RYR1*	R A B	Het	rs2145586397	Chr19(GRCh37):g.38987095G>A; NM_000540.3:c.6710G>A (p.Cys2237Tyr)
9	*RYR1*	H B O	Het	rs748318655 rs1434232361	Chr19(GRCh37):g.38959643G>A; NM_000540.3:c.3419G>A (p.Arg1140His) Chr19(GRCh37):g.39019006C>T; NM_000540.3:c.10885C>T (p.Arg3629Trp)
10	*RYR1*	H B	Het	rs771320029	Chr19(GRCh37):g.39017687G>A; NM_000540.3:c.10681G>A (p.Gly3561Arg)

Hemi = hemizygous, Het = heterozygous.

## Data Availability

The data supporting the findings of this study are available on request from the corresponding author. The data are not publicly available due to privacy or ethical restrictions.

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
