# Peer review of "Rhabdomyolysis in the Era of Next-Generation Sequencing: Selecting Patients with a High Likelihood of a Genetic Susceptibility Using ‘RHABDO’ Features"

_genes, 2025, doi:10.3390/genes16090995_

Round 1

Reviewer 1 Report

Comments and Suggestions for Authors

The authors in this manuscript performed a retrospective study on the patients of rhabdomyolysis. The authors performed the Whole Exome Sequencing in the known patients of rhabdomyolysis and developed a RHABDO tool which will suggest whether a patient should go for genetic testing or not. The study is interesting and useful for the community. The manuscript is well written, however there is some typo (line 83 and lines 161, 162, 169 'pathogenic or pathogenic variants'). The conclusions drawn are supported by the result. My comments are provided below

  1. The authors need to discuss the Table 1. The p value for some of the RHABDO features arenot significant. Is this the difference between the method of sequencing groups or But later on the p value was shown 0.03. The authors need to provide more explanation this.
  2. The limitations of the study needs to be discussed. The study wasnot performed on the unknown patients.  
  3. Table 2 should also include the number of patients for each variant.

Author Response

Please find the rebuttal letter in the attached zip file.

Reviewer 2 Report

Comments and Suggestions for Authors

The authors conducted a multicenter retrospective study involving 122 patients in the Netherlands who underwent genetic testing since NGS introduction in 2013, aimed to evaluate the diagnostic approach and the predictive value of RHABDO features. A key update highlighted by the research is the importance of reanalyzing patients with unexplained rhabdomyolysis using updated NGS gene panels, as the number of tested genes continuously expands. The reviewer agrees that an international, collaborative approach with standardized data collection is essential to further evaluate RHABDO genetic testing and its interpretation in clinical settings. Overall, this is an excellent retrospective study that will be useful for future rhabdomyolysis studies.

Minor comments

  1. Figure 1 is missing.
  2. It will be informative to give the number of patients harboring a specific gene variant in Table 3, for example, # of patients carrying the variant of Chr17(GRCh37):g.7125494A>C in the ACADVL gene. The rsID should also be listed.

Author Response

(The authors gave the same response as above.)

Reviewer 3 Report

Comments and Suggestions for Authors

The present manuscript describes a descriptive study aimed at evaluating whether specific clinical features of rhabdomyolysis could help prioritize patients for genetic testing, particularly whole-exome sequencing (WES). The study is methodologically sound, and the authors have appropriately acknowledged its main limitations.

However, several aspects would benefit from further clarification. For this reason, I suggest to deeply revise the Discussion:

-The reported positive predictive value is 14%. It is unclear whether such a value meaningfully supports the authors' conclusions. Could this PPV be improved by employing broader genomic approaches such as whole-genome sequencing (WGS)? Alternatively, might the inclusion of large copy number variant (CNV) analyses yield a higher diagnostic yield?

The utility of the "RHABDO" features in guiding genetic testing should be more thoroughly explained. Specifically, in what proportion of cases do these features improve diagnostic accuracy or influence clinical management compared to standard evaluation approaches for rhabdomyolysis and subsequent genetic testing?

Future perspectives of the study should be better defined. For example, regarding patients with variants of uncertain significance (VUS), do the authors plan to follow these individuals longitudinally to assess whether changes in RHABDO features may help refine the interpretation of such variants?

Finally, given that only two centers contributed to the study, I recommend removing the term "multicentric" from the title and throughout the manuscript, as it may overstate the scope of the cohort.

Author Response

Please find the rebuttal letter in the attached PDF file.

Round 2

Reviewer 3 Report

Comments and Suggestions for Authors

The manuscript has improved following suggested revision